A new species of Escallonia (Escalloniaceae) from the inter-Andean tropical dry forests of Bolivia

Zapata Felipe fzapata@ucla.edu 1
Villarroel Daniel 2 3
1 Department of Ecology and Evolutionary Biology, University of California, Los Angeles , CA , United States of America
2 Fundación Amigos de la Naturaleza , Santa Cruz , Bolivia
3 Universidad Autónoma Gabriel René, Museo de Historia Natural Noel Kempff Mercado , Santa Cruz , Bolivia
Sosa Victoria
Electronic publication date: 2019 Jan 31
Publication date: 2019
Volume: 7
Electronic Location ID: e6328
Received 2018 Sep 27; Accepted 2018 Dec 21
Copyright: ©2019 Zapata and Villarroel
Copyright year: 2019
Copyright holder: Zapata and Villarroel
License: This is an open access article distributed under the terms of the Creative Commons Attribution License, which permits unrestricted use, distribution, reproduction and adaptation in any medium and for any purpose provided that it is properly attributed. For attribution, the original author(s), title, publication source (PeerJ) and either DOI or URL of the article must be cited.
License URL: https://creativecommons.org/licenses/by/4.0/

Keywords: Morphometrics, Species limits, Dry forest, Escalloniaceae, Neotropics, Taxonomy

Funding: CAPES (Coordenação de Aperfeiçoamento de Pessoal de Nível Superior) Universidade de Brasilia Daniel Villarroel received support from CAPES (Coordenação de Aperfeiçoamento de Pessoal de Nível Superior) and the Universidade de Brasilia. The funders had no role in study design, data collection and analysis, decision to publish, or preparation of the manuscript.

==============================
Over the last two decades, renewed fieldwork in poorly explored areas of the tropical Andes has dramatically increased the comparative material available to study patterns of inter- and intraspecific variation in tropical plants. In the course of a comprehensive study of the genus Escallonia, we found a group of specimens with decumbent branching, small narrowly elliptic leaves, inflorescences with up to three flowers, and flowers with red petals. This unique combination of traits was not present in any known species of the genus. To evaluate the hypothesis that these specimens belonged to a new species, we assessed whether morphological variation between the putative new species and all currently known Escallonia species was discontinuous. The lack of overlap in tolerance regions for vegetative and reproductive traits combined with differences in habit, habitat, and geographic distribution supported the hypothesis of the new species, which we named Escallonia harrisii. The new species grows in sandstone inter-Andean ridges and cliffs covered with dry forest, mostly on steep slopes between 1,300–2,200 m in southern Bolivia. It is readily distinct in overall leaf and flower morphology from other Escallonia species in the region, even though it does not grow in sympatry with other species. Because E. harrisii is locally common it may not be threated at present, but due to its restricted geographic distribution and the multiple threats of the tropical dry forests it could become potentially vulnerable.

Introduction

The tropical Andes harbor an exceptional concentration of endemic plant species and are considered one of the hottest global biodiversity hotspots (Myers et al., 2000). Patterns of species richness and endemicity in these mountains vary with elevation as a result of the evolutionary history of resident lineages. The low elevation tropical dry forest stands out as a remarkable biome that includes more species-poor but endemic-rich clades than other Andean biomes due to the persistence of old lineages that have diversified over the last 20 my (Särkinen et al., 2011). Unfortunately, these evolutionarily unique forests are highly threatened (Banda-R et al., 2016). Therefore discovering, describing, and documenting their biodiversity is of significant interest to evolutionary and conservation biologists alike.

Escallonia Mutis ex L.f. (Escalloniaceae) is a morphologically and ecologically diverse genus of shrubs and small trees widely distributed in the neotropical mountains (Zapata, 2013; Sede & Denham, 2018). It is characterized by its sympodial growth with distinctive long- and short-shoot construction. The leaves are always simple, spiral, and with serrate margins. Flowers are borne singly or in inflorescences of few to many flowers. The flowers are always pentamerous, with free petals at maturity and inferior ovaries. There is an intrastaminal nectary disk, and always a characteristic large discoid stigma. All Escallonia species have bilocular septicidal capsules enclosing about 100 minute seeds. The species examined so far show the same chromosome morphology and base number (n = 12) (Zielinski, 1955; Sanders, Stuessy & Rodriguez, 1983; Hanson et al., 2003). Some species have a long history in horticulture and are widely used as ornamentals (Sleumer, 1968; Denaeghel et al., 2018).

Escallonia, with 39 species, is one the most species-rich genera in the Escalloniaceae (APG, 2016). Although relationships within Escalloniaceae and between Escalloniaceae and other Campanulids are not fully resolved (Tank & Donoghue, 2010; Beaulieu & O’Meara, 2018), the monophyly of Escallonia is strongly supported (Sede et al., 2013; Zapata, 2013). Most Escallonia species are distributed along the Andes, from northern Venezuela to southern Argentina, and the mountains of Costa Rica. Some species are restricted to the mountains of southeastern Brazil, and one species occurs in Juan Fernández Island. Most species have comparatively broad geographic ranges, and only few species are extremely narrow endemics (Sleumer, 1968). The phylogeny of Escallonia shows considerable phylogenetic geographic structure with major clades restricted to geographic regions (Zapata, 2013). This suggests that old divergences are associated with geographic isolation and that recent divergences are associated with bioclimatic differentiation along elevation gradients within geographic regions.

Historically, Escallonia has been relatively well-collected in some areas such as the southern Andes (Kausel, 1953; Sleumer, 1968; Sede & Denham, 2018). Renewed field exploration in poorly-known and highly threatened regions of the tropical Andes has made available new comparative material to study broad patterns of variation and reassess species boundaries in the genus. In this study, we present and describe a new species of Escallonia restricted to the dry forests of southern Bolivia. We include a detailed description and an illustration, and a discussion of the eco-phenotypic differences between the new species and other species that occur in the region.

Materials & Methods

Species concept

In the present study, we follow the general lineage species concept (De Queiroz, 1998), which proposes that species are independently evolving segments of population-level lineages and that any evidence of lineage separation (i.e., distinct morphology, differences in ecological niche, monophyly of alleles) is sufficient to infer the existence of separate species (De Queiroz, 2007). Here, we assess discontinuities in continuous morphological traits using the approach proposed by Zapata & Jiménez (2012), in combination with differences in habit, habitat and geographic distribution.

Taxon sampling

A total of 809 herbarium specimens from all species of Escallonia and the new species were included in this study. Escallonia salicifolia Mattf. was not included here because only two specimens were available and the method used in this study requires a sample size larger than three (Zapata & Jiménez, 2012). Voucher information for all specimens is available in a Git repository at http://github.com/zapataf/ms_eharrisii.

Morphological measurements

Because the new species differs from other Escallonia species in overall leaf shape and flower number, we tested the hypothesis that the new species boundary exists in the morphological spaces defined by these traits. To quantify leaf shape, we measured leaf length and width, and we counted the number of flowers per inflorescence in all specimens. On each specimen, we recorded leaf measurements from three different leaves and then averaged to generate mean leaf measurements. We counted flower number on one inflorescence per specimen.

Morphological discontinuities

We assessed morphological discontinuities in leaf shape and flower number between the new species and all Escallonia species using the method of Zapata & Jiménez (2012). This pairwise method assesses whether the overlap of morphological values is below a given threshold to indicate a discontinuity, and thus support a species boundary. To determine morphological overlap, this method estimates the overlap of statistical tolerance regions in the morphological space defined by measurements of two species. Statistical tolerance regions correspond to the regions estimated with statistical confidence γ that encompass a proportion β of a population (i.e., a species). In this study, we used statistical confidence γ = 0.90 and a threshold 0.15. Therefore, we inferred a morphological discontinuity when proportions β ≥ 0.85 (one minus threshold) of each species in a pairwise comparison did not overlap. For leaf shape, we estimated the statistical tolerance regions for each species sharing a single point along the ridgeline manifold. The ridgeline manifold is the curve that includes all the critical points (maxima, minima, and saddles) of the bivariate normal mixture describing variation in leaf shape implied by each pairwise comparison. For flower number, we estimated one-tailed statistical tolerance regions describing variation in flower number implied by each pairwise comparison. All analyses were carried out in R 3.5.0 (R Core Team, 2016); source code and the data used in these analyses are available in a Git repository at http://github.com/zapataf/ms_eharrisii.

Nomenclature

The electronic version of this article in Portable Document Format (PDF) will represent a published work according to the International Code of Nomenclature for algae, fungi, and plants (ICN), and hence the new names contained in the electronic version are effectively published under that Code from the electronic edition alone. In addition, new names contained in this work which have been issued with identifiers by IPNI will eventually be made available to the Global Names Index. The IPNI LSIDs can be resolved and the associated information viewed through any standard web browser by appending the LSID contained in this publication to the prefix “http://ipni.org/”. The online version of this work is archived and available from the following digital repositories: PeerJ, PubMed Central, and CLOCKSS.

Results & Discussion

Discontinuity in leaf morphology

The new species has small narrowly elliptic leaves (14.5–19.7 × 1.6–3.7 mm), which are uncommon in Escallonia (Fig. 1). The weight of the evidence supporting a morphological discontinuity in leaf morphology is strong between the new species and the following 20 species: E. angustifolia C. Presl, E. bifida Link & Otto, E. chlorophylla Cham. & Schltdl., E. farinacea A. St.-Hil., E. herrerae Mattf., E. hispida (Vell.) Sleumer, E. illinita C. Presl, E. laevis (Vell.) Sleumer, E. micrantha Mattf., E. millegrana Griseb., E. myrtoidea Bertero ex DC., E. obtusissima A. St.-Hil., E. paniculata (Ruiz & Pav.) Roem. & Schult., E. pendula (Ruiz & Pav.) Pers., E. petrophila Rambo & Sleumer, E. piurensis Mattf., E. pulverulenta (Ruiz & Pav.) Pers., E. reticulata Sleumer, E. revoluta (Ruiz & Pav.) Pers., and E. schreiteri Sleumer (Fig. 2). These results support the hypothesis of a species boundary between the new species and 20 currently known Escallonia species.

Figure 1 Variation in leaf morphology in Escallonia.

Each panel corresponds to the pairwise comparison between a currently known species (dark circles) and the new species (E. harrisii; light triangles) for lamina length (X-axis) and lamina width (Y-axis), E. harrisii vs. A, E. alpina; B, E. angustifolia; C, E. bifida; D, E. callcottiae; E, E. chlorophylla; F, E. cordobensis; G, E. discolor; H, E. farinacea; I, E. florida; J, E. gayana; K, E. herrerae; L, E. hispida; M, E. hypoglauca; N, E. illinita; O, E. laevis; P, E. ledifolia; Q, E. leucantha; R, E. megapotamica; S, E. micrantha; T, E. millegrana; U, E. myrtilloides; V, E. myrtoidea; W, E. obtusissima; X, E. paniculata; Y, E. pendula; Z, E. petrophila; AA, E. piurensis; BB, E. polifolia; CC, E. pulverulenta; DD, E. resinosa; EE, E. reticulata; FF, E. revoluta; GG, E. rosea; HH, E. rubra; II, E. schreiteri; JJ, E. serrata; KK, E. tucumanensis; LL, E. virgata.

Figure 2 Discontinuity in leaf morphology in Escallonia.

Each panel corresponds to the pairwise comparison between a currently known species (dark continuous line) and the new species (E. harrisii; light continuous line). The continuous lines show the estimated proportion (Y-axis) covered with confidence γ = 0.90 by the tolerance regions sharing a single point along the ridgeline manifold (X-axis). The ridgeline manifold is the curve that includes all the critical points of the bivariate normal mixture describing the morphological variation implied by each pairwise comparison in Fig. 1. The ridgeline manifold ranges from zero, the bivariate mean of the new species (E. harrisii), to one, the bivariate mean of the currently known species. The dashed line marks proportion β = 0.85 (one minus threshold 0.15) below which overlap indicates lack of a morphological discontinuity. E. harrisii vs. A, E. alpina; B, E. angustifolia; C, E. bifida; D, E. callcottiae; E, E. chlorophylla; F, E. cordobensis; G, E. discolor; H, E. farinacea; I, E. florida; J, E. gayana; K, E. herrerae; L, E. hispida; M, E. hypoglauca; N, E. illinita; O, E. laevis; P, E. ledifolia; Q, E. leucantha; R, E. megapotamica; S, E. micrantha; T, E. millegrana; U, E. myrtilloides; V, E. myrtoidea; W, E. obtusissima; X, E. paniculata; Y, E. pendula; Z, E. petrophila; AA, E. piurensis; BB, E. polifolia; CC, E. pulverulenta; DD, E. resinosa; EE, E. reticulata; FF, E. revoluta; GG, E. rosea; HH, E. rubra; II, E. schreiteri; JJ, E. serrata; KK, E. tucumanensis; LL. E. virgata.

Discontinuity in flower number

The new species has inflorescences with up to three flowers, which are uncommon in Escallonia (Fig. 3). There is support for a morphological discontinuity in flower number between the new species and 30 species, 19 of which are also separated by a discontinuity in leaf morphology (see above, all species except E. petrophila). The remaining 11 species separated only by a morphological discontinuity in flower number are: E. alpina Poepp. ex DC., E. cordobensis (Kuntze) Hosseus, E. discolor Vent., E. florida Poepp. ex DC., E. hypoglauca Herzog, E. leucantha J. Rémy, E. megapotamica Spreng., E. resinosa (Ruiz & Pav.) Pers, E. rubra (Ruiz & Pav.) Pers., E. tucumanensis Hosseus, and E. virgata (Ruiz & Pav.) Pers. (Fig. 4). These results support the hypothesis of a species boundary between the new species and 30 currently known Escallonia species.

Figure 3 Variation in flower number in Escallonia. New species (E. harrisii) at the left.

Figure 4 Discontinuity in flower number in Escallonia.

Each panel corresponds to the pairwise comparison between a currently known species (dark color) and the new species (E. harrisii; light color). The dotted and dashed lines estimate with confidence γ = 0.90 the limits of one-tailed tolerance regions encompassing a proportion β = 0.85 (one minus threshold 0.15) of the currently known and new species, respectively. E. harrisii vs. A, E. alpina; B, E. angustifolia; C, E. bifida; D, E. callcottiae; E, E. chlorophylla; F, E. cordobensis; G, E. discolor; H, E. farinacea; I, E. florida; J, E. gayana; K, E. herrerae; L, E. hispida; M, E. hypoglauca; N, E. illinita; O, E. laevis; P, E. ledifolia; Q, E. leucantha; R, E. megapotamica; S, E. micrantha; T, E. millegrana; U, E. myrtilloides; V, E. myrtoidea; W, E. obtusissima; X, E. paniculata; Y, E. pendula; Z, E. petrophila; AA, E. piurensis; BB, E. polifolia; CC, E. pulverulenta; DD, E. resinosa; EE, E. reticulata; FF, E. revoluta; GG, E. rosea; HH, E. rubra; II, E. schreiteri; JJ, E. serrata; KK, E. tucumanensis; LL, E. virgata.

Support for the new species boundary

Taken together, the results described above show there is evidence supporting a species boundary between the new species and 31 currently known Escallonia species. In most cases the new species boundary spans differences in both leaf and flower traits (19 species). In other cases, the species boundary is supported with evidence from one of the traits (12 species). This is consistent with the species concept we apply in this study (De Queiroz, 2007). For instance, E. micrantha and the new species differ in both leaf shape and flower number, whereas E. florida and the new species differ in flower number but are broadly similar in leaf shape (Figs. 2, 4). No material for DNA sequencing was available to place the new species in the Escallonia phylogeny, therefore it is not possible to discern whether morphological similarities between the new species and other species reflect convergent evolution or recent divergence with little differentiation.

Lack of support for morphological discontinuities, differences in habit and habitat, and alternative explanations

The weight of the evidence supporting a morphological discontinuity between the new species and the following seven species was weak: E. callcottiae Hook. & Arn., E. gayana Acevedo & Kausel, E. ledifolia Sleumer, E. myrtilloides L. f., E. polifolia Hook., E. rosea Griseb., and E. serrata Sm. There are three non-exclusive reasons to explain why this result does not undermine the hypothesis of a species boundary between the new species and any of these seven species: (i) Habit and habitat. E. myrtilloides, E. polifolia, E. rosea, E. serrata, and the new species all differ in habit and habitat. E. myrtilloides includes small trees with thick branches, obovate glabrous leaves, and it is restricted to the páramos and jalcas in the tropical Andes above 2,600 m. E. polifolia includes small shrubs with revolute, tomentulose leaves, and it is endemic to the jalcas in the Cha-Chapoyas region (northern Perú) above 2,800 m. E. rosea includes shrubs with lanceolate to obovate-lanceolate leaves, largely glabrous (only pubescent adaxially along the mid vein), and it is restricted to the wet temperate forests of southern Chile (Valdivian forests). E. serrata includes procumbet shrubs with mostly obovate-cuneate leaves, often entirely glabrous (sometimes puberulous adaxially along the veins), and it is endemic to Patagonia in southern Chile and Argentina. In contrast, the new species includes profusely branched sub-shrubs with decumbent branching and slender twigs, and it is endemic to the dry forest in southern Bolivia at around 1,700 m. (ii) Geographic sampling. One could propose that the new species is an allopatric population of any of the seven species. This would predict there are unsampled populations from any of the seven species across the geographic range of Escallonia. This is highly unlikely because other Escallonia species have been sampled thoroughly at intervening localities (Fig. 5) and we have examined around 3,900 Escallonia herbarium specimens that indicate that the geographic range of the seven species is well sampled (Zapata, 2011). Lastly, (iii) Statistical power. The sample size for E. callcottiae, E. gayana, and E. ledifolia is very low (Table 1), which lowers the statistical power of the method we used to diagnose morphological discontinuities (Zapata & Jiménez, 2012). Therefore the lack of evidence supporting a morphological discontinuity in these cases may just be a statistical artifact.

Figure 5 (A) Geographic distribution of species lacking support for morphological discontinuities vs the new species (E. harrisii). Grey points correspond to all other Escallonia species. (B) Elevation range of species lacking support for morphological discontinuities vs. the new species (E. harrisii).

Table 1 Descriptive statistics.

	N	mLL	MeLL	MLL	mLW	MeLW	MLW	mFN	MeFN	MFN	mE	ME	
E. alpina	40	9.2	16.6	26.5	4	7.0	9.8	1	8.1	14	20	2,300	
E. angustifolia	13	39.5	54.6	74	5	11	17	16	70.3	155	1,600	3,280	
E. bifida	36	41.7	55.9	79	13.3	17.4	23.3	25	83.1	150	70	2,300	
E. callcottiae	6	21.7	29.2	42.3	9.7	13	21.7	12	23.7	55	40	800	
E. chlorophylla	13	37.3	47.6	56.7	12.6	17.6	23.4	27	42.1	60	0	1,312	
E. cordobensis	11	29.7	40.6	53.5	6	7.9	10.9	7	12.7	22	1,000	2,400	
E. discolor	5	48.7	59.0	76.7	17.7	20.9	23.3	65	103	150	2,500	3,300	
E. farinacea	18	35	48.6	62.3	10	14.6	18.3	6	13.9	22	812	1,810	
E. florida	10	16	18.8	23	3	3.7	6	10	31.3	61	624	2,000	
E. gayana	7	13.3	16.9	21.7	4.3	5.6	7.1	28	48	100	100	800	
E. harrisii	6	14.5	16.8	19.7	1.6	2.8	3.7	1	1.8	3	1,350	2,200	
E. herrerae	8	103.7	152.1	174.3	24.3	35	46	115	157	250	1,800	3,450	
E. hispida	8	35	47.2	55.8	16.2	20	22.2	16	20.3	34	600	1,500	
aE. hypoglauca	24	18	30.3	48.3	7.3	13.3	22.3	7	10.9	24	2,200	3,500	
E. illinita	25	37.7	45.8	58.7	8.3	16.4	21.7	22	49.5	150	40	2,650	
E. laevis	27	20.3	37	57.4	9.6	13.8	21	7	14	30	0	2,750	
E. ledifolia	10	27.7	35.9	43	5	7.4	9.7	2	5.6	8	950	1,150	
E. leucantha	14	15.7	22.7	31	5.3	8.3	11	33	61.2	80	50	707	
E. megapotamica	29	21	31.2	48	5.5	8.9	16	19	49.11	95	30	1,000	
E. micrantha	11	88.3	118.8	142.3	23	32.4	43.3	800	881.8	950	1,850	2,500	
aE. millegrana	21	73.3	104.8	146.3	22.5	32.5	46	700	835.7	950	1,228	2,950	
aE. myrtilloides	56	6.7	14.4	30.3	3.8	7.1	18	1	1	1	2351	4,100	
E. myrtoidea	15	31.7	45	72	15.3	20.9	26	35	73.8	115	120	2,000	
E. obtusissima	6	52	61.2	69.3	19	22.7	27.7	27	32.8	45	800	1,200	
aE. paniculata	66	40.7	75.7	112	16.3	24	37	40	148.9	800	1,200	3,492	
E. pendula	30	116.7	175.4	217.7	22.3	38.3	55.7	80	169.3	280	1,300	3,100	
E. petrophila	8	55.8	84.6	98.7	20.9	25.9	29	5	7	11	800	1,131	
E. piurensis	12	24	36.5	44.7	9.7	11.4	13.7	20	44.9	75	2,500	3,300	
E. polifolia	9	13.8	17.6	19.5	2.3	2.8	3.8	1	1	1	2,900	3,500	
E. pulverulenta	25	41.9	54.6	72.9	17.3	26.5	37	140	171.5	200	0	1,200	
aE. resinosa	46	20.7	34.1	52.7	5.7	8.6	13	10	56.4	130	2,200	3,776	
aE. reticulata	21	55.7	68.7	80	15	23.7	28.3	25	55.8	100	1,300	2,400	
E. revoluta	21	31.7	39.1	49.3	12.8	18.7	25.7	27	75.3	180	1	1,642	
E. rosea	30	15	33.5	51	5	12.8	24.3	5	16.7	55	185	1,662	
E. rubra	46	21.7	36.6	58.3	6.3	16.7	32	5	15.5	45	0	1,605	
aE. schreiteri	18	42.3	56.6	74	8.3	11.1	14.7	20	42.6	60	1,600	2,954	
E. serrata	20	8.2	14.4	21.2	4.3	6.6	9.5	1	1	1	5	400	
aE. tucumanensis	18	31.7	50.5	76	11	18.3	30.7	6	14.2	30	800	2,800	
E. virgata	20	9.5	11.7	14.7	3.5	4.5	5.7	6	13.1	28	61	3,000	
Notes.

N sample size

m minimum

Me mean

M maximum

LL leaf length

LW leaf width

FN flower number

E elevation

New species in bold.

a Species occurring in Bolivia.

Taxonomic Treatment

Escallonia harrisii F Zapata & Villarroel, sp. nov. (Fig. 6)

Figure 6 Escallonia harrisii.

F Zapata & Villarroel. a. Habit, b. Fruit, c. Flower, d. Flower with petals removed, e. Petal, f. Mature leaf, g. Young leaf, h. Detail of leaf margin, i. Detail of outer bark in mature shoot. Illustration by B. Alongi based on Vargas, I.G. 3673 (MO) and Wood et al. 13266 (K).

Type: BOLIVIA. Santa Cruz. Vallegrande. San Blas (abajo) y La Estancilla, 5–8 km de la ciudad de Vallegrande. 18°29′S, 63°59′W, 2,200 m, 19 November 1994 (fl), Vargas, I.G. 3673 (holotype: MO, isotypes: NY; USZ).

Paratype: BOLIVIA. Chuquisaca. 4 km de la comunidad de San Bartolo, sobre el camino a Nuevo Mundo, 19°39′S, 64°02′W, 1,350 m, 14 December 2013 (fl, fr), Villarroel et al. 2322 (UB; USZ).

Diagnosis: Decumbent branching, small narrowly elliptic leaves (14.5–19.7 × 1.6–3.7 mm), inflorescences with up to three flowers, flowers with red petals.

Description: Perennial sub-shrub, to 2 m tall, profusely branched, branches decumbent, 1.9–1.0 mm diameter, angular to terete, outer bark scaly, grey, new growth branches angular, outer bark smooth, reddish, densely puberulent, hairs simple, white, 0.1–0.2 mm long. Leaves spiral; petiole 0.6–0.8 mm long; lamina oblanceolate, 14.5–19.7 × 1.6–3.7 mm, basally attenuate, apically acute, abaxially dull with scattered glands, minutely puberulent (simple hairs), adaxially lustrous green, glabrous; margin slightly serrate, glandular; secondary veins three to four pairs, brochidodromous. Inflorescences terminal, 1 to 3-flowered. Flowers hermaphrodite, pentamerous. Pedicels 2–4.7 mm long, 0.4–0.6 mm diameter, terete, densely puberulent (simple hairs). Ovary inferior, turbinate, 1.5–3 × 2.4–3.8 mm, puberulent. Calyx tube 0.5–0.7 mm long; lobes narrowly triangular-subulate, 5.8–10 × 0.8–1.3 mm, abaxially and adaxially puberulent, margin glandular, sparsely ciliate, sometimes recurved. Corolla actinomorphic, glabrous; petals red, spatulate, 6.7–7.9 mm long, 0.9–1.17 mm wide at base, 1.5–1.75 mm at the widest point, margin minutely crenulate. Stamens 5; filaments glabrous, terete, 4.2–4.9 mm long; anthers versatile, sub-basifixed, narrowly oblong 1.30 × 0.44 mm. Style terete, 4.4–6.3 mm long. Stigma discoid. Disk flat. Fruit brown, turbinate, 3.10–4.12 × 3.41–4.70 mm, dehiscence septicidal. Seeds linear, 0.04 mm long, striate.

Additional Specimens Examined: BOLIVIA. Chuquisaca. Calvo. Serranía Incahuasi. 10–15 km from Muyupampa on road to Lagunillas, 19°21′51″S, 63°50′09″W, 1,500 m, 8 March 1998 (fl, fr), Wood et al. 13266 (K); Chuquisaca. Calvo. Serranía del Incahuasi, 10–15 km de Muyupampa sobre el camino a Lagunillas, 19°49′39″S, 63°43′31″W, 1,580 m, 25 March 2013 (fl, fr), Wood et al. 27640 (K; USZ); Chuquisaca. Calvo. Serranía Incahuasi, entre Muyupampa y Lagunillas, 19°49′38″S, 63°43′30″W, 1,580 m, 13 December 2013 (fl, fr), Villarroel et al. 2321 (UB; USZ); Tarija. O’Connor. On w side of easternmost pass on road from Entre Rios to Palos Blancos, 21°25′28″S, 63°54′47″W, 1,400 m, 17 January 2001 (fr), Wood and Goyder 16822 (K).

Etymology: The specific epithet is in honor of Whitney R. Harris, who supported the center that now bears his name, the Whitney R. Harris World Ecology Center at the University of Missouri-St. Louis. Through the support provided by this center, several generations of biologists from throughout the world have been able to contribute to the study, understanding, and conservation of temperate and tropical ecosystems worldwide.

Phenology: Flowering and fruiting specimens have been collected between November and March. There are no observations yet on pollination or dispersal biology.

Distribution: Restricted to the south of Bolivia (Fig. 7). Locally common.

Figure 7 (A) Collection sites of the new species E. harrisii. (B) Map of South America, Bolivia shaded. (C) Map of Bolivia, area of A. shaded. (Map Credit: ©2018 Google, TerraMetrics).

Habitat: Plants of this species grow on rocky outcrops and ridges of red sandstone, mostly on steep slopes and summits between 1,300–2,200 m elevation. The dominant vegetation in the region where E. harrisii grows is dry forest on the slopes (i.e., Chaco Serrano forest) and semi-deciduous forests on the mountaintops (i.e., Tucumano Boliviano forest)

Conservation status: Although E. harrisii has been collected in few localities, it is locally common and it may not be threatened at present. Because it is restricted to the tropical dry forest, one of the most threatened tropical habits (Banda-R et al., 2016), it could become potentially vulnerable. However, more data and population-level studies are needed to assess the conservation status of this species. An IUCN category of Data Deficient (DD) is assigned, according to IUCN criteria (IUCN, 2012).

Affinities: Because we did not have access to good quality DNA, E. harrisii has not been included in a molecular phylogenetic study of Escallonia and its closest relatives are not known. Morphologically, E. harrisii displays similarities in leaf shape and flower number with E. callcottiae, E. gayana, E. ledifolia, E. myrtilloides, E. polifolia, E. rosea and E. serrata (Table 1). However, none of these species has decumbent branching, slender twigs and narrow oblanceolate leaves.

Ecologically, no other species of Escallonia has been found in sympatry with E. harrisii. Only E. millegrana, E. micrantha, E. pendula and E. herrerae grow at equivalent elevations and in similar habitats (i.e., dry forests in inter-Andean valleys). These four species are strikingly different in all morphological traits compared to E. harrisii (Table 1). Although E. millegrana also occurs in Bolivia, plants of this species are tall deciduous shrubs (up to 4 m) with long leaves (up to 15 cm), spines in young shoots, and inflorescences with around 850 flowers (Table 1).

Key to the Escallonia species in the region where E. harrisii occurs

1. Calyx lobes > 1 mm	2	
2. Petal length < 5 mm	E. millegrana	
2a. Petal length ≥ 5 mm	3	
3. Pedicel width > 0.7 mm (stout, rigid)	E. myrtilloides	
3a. Pedicel width < 0.5 mm	4	
4. Lamina width at the widest point 11–30 mm; petal length 13–17 mm	E. tucumanensis	
4a. Lamina width at the widest point < 22 mm; petal length < 12 mm	5	
5. Lamina width at the widest point < 4 mm	E. harrisii	
5a. Lamina width at the widest point > 7 mm	E. hypoglauca	
1a. Calyx lobes < 0.5 mm	6	
6. Petiole length < 10 mm; lamina width at the widest point < 15 mm	7	
7. Lamina length < 50 mm, oblongo-cuneate; petal length ≤ 6 mm	E. resinosa	
7a. Lamina length > 42 mm, lanceolate; petal length > 6 mm	E. schreiteri	
6a. Petiole length ≥ 10 mm; lamina width at the widest point ≥ 15 mm	8	
8. Pedicel length 4–6 mm; restricted to dry forests	E. reticulata	
8a. Pedicel length 2–3 mm; restricted to cloud forests	E. paniculata	

We thank Peter F. Stevens and Elizabeth A. Kellogg for their constructive suggestions and valuable comments on earlier versions of the manuscript. We also thank the curators and collection managers of the herbaria cited for the use of their specimens, in particular to James Solomon (MO). Thanks to CAPES (Coordenaão de Aperfeioamento de Pessoal de Nivel Superior) and the University of Brasilia (UNB) for a doctoral scholarship to DV. We are grateful to Barbara Alongi for preparing the beautiful illustration of E. harrisii.

Additional Information and Declarations

Competing Interests

Author Contributions

Data Availability

New Species Registration

The authors declare there are no competing interests.

Felipe Zapata conceived and designed the experiments, performed the experiments, analyzed the data, contributed reagents/materials/analysis tools, prepared figures and/or tables, authored or reviewed drafts of the paper, approved the final draft.

Daniel Villarroel performed the experiments, contributed reagents/materials/analysis tools, authored or reviewed drafts of the paper, approved the final draft.

The following information was supplied regarding data availability:

Data is available at GitHub: https://github.com/zapataf/ms_eharrisii.

The following information was supplied regarding the registration of a newly described species:

Escallonia harrisii F Zapata & Villarroel, LSID: 77193060-1.

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
