# Peer review of "A new species of Escallonia (Escalloniaceae) from the inter-Andean tropical dry forests of Bolivia"

_PeerJ, doi:10.7717/peerj.6328_

## Round 0.1 · original submission · Minor Revisions

Both reviewers suggested only minor changes and agreed that this paper is well written and that analyses to delimit species are novel and very interesting. Please pay attention to comments of Reviewer 1 in relation to clarify selection of characters used in analyses on one hand and on the other the inclusion of a key to recognize this new species with species distributed in Bolivia and the north of Argentina.

·

Basic reporting

no comment

Experimental design

no comment

Validity of the findings

no comment

Additional comments

I have had the pleasure to review the manuscript ‘A new species of Escallonia (Escalloniaceae) from the inter-Andean tropical dry forests of Bolivia’.

The major highlights of this contribution are the number of specimens analyzed and the interesting statistical approach they have employed to find morphological discontinuities among species and hence hypothesize species limits. The manuscript is well written, well referenced and results are supported by the analysis and discussed correctly.

I have a few suggestions in order to improve this contribution:
1- Materials and Methods
Morphological measurements
I think authors should clarify why they chose leaf length and width and the number of flowers from among 6 vegetative and 21 reproductive characters used by Zapata & Jiménez (2012).

2- Results
Support for the new species boundary
Taking into account that the published phylogenies show high support for a geographical structuring of the groups, I would add a phrase hypothesizing about the probable position of E. harrisii.

Lack of support for morphological discontinuities, differences in habit and habitat, and alternative explanations
In the same way you added the characters “revolute, tomentulose leaves” to E. polifolia, I think you can add other relevant characters to differentiate the remaing species.

After “Taxonomic treatment”, and to conclude, I suggest including a short key to differentiate E. harrisii from the remaining species of the region (Bolivia and northern Argentina), using discontinuities in leaf morphology and flower number, and also some other characters / habitat, etc…

Finally, I have annotated the pdf file; place special attention to Table and Figure legends.
It was difficult to get to the additional information; please take a look at the links.

Reviewer 2 ·

Basic reporting

'no comment'

Experimental design

'no comment'

Validity of the findings

'no comment'

Additional comments

The present paper describes a new species of plants from Bolivia. Authors incorporate recent methods for species discrimination to validate their proposal. This study is thorough, scientifically, and exemplar; to my knowledge is one of the best species descriptions out in the literature because it incorporates statistical comparison with other species in the genus. I believe this study would become an example of great contemporary taxonomy, reason why the timely publication of this study is urgent.

The study is clear, professional, and well written from title to references. The introduction is appropriate, methods are well thought, results are well explained, and the interpretations are logical. I praise that authors identify and explain the weaknesses of their analyses.

Things to improve:
- Authors should mention the new species name in all the figures, that way every figure is completely self-explained and understandable.
- Proportion plots (fig 2) are neither explained in the text nor in the figure. As a reader I was not able to interpret those. An explanation of these plots should be included in the text.
- See minor edits in the pdf attached.

Annotated reviews are not available for download in order to protect the identity of reviewers who chose to remain anonymous.

---

## Round 0.2 · accepted · Accept

Thank you for considering the suggestions by the two reviewers, I am happy to Accept the paper. My only concern is that according to the International Plant Names Index, the author name Zapata has been given to Mario Zapata, thus I recommend that you use the initial of your name.

#